# Enhancement of fear memory by retrieval through reconsolidation

**Hotaka Fukushima[1,2,†], Yue Zhang[1,2,†], Georgia Archbold[3], Rie Ishikawa[1], Karim Nader[3], Satoshi Kida[1,2]\***

[1]Department of Biosciences, Faculty of Applied Bioscience, Tokyo University of Agriculture, Tokyo, Japan; [2]Core Research for Evolutionary Science and Technology (CREST), Japan Science and Technology Agency, Saitama, Japan; [3]Department of Psychology, McGill University, Montreal, Canada

**Abstract** Memory retrieval is considered to have roles in memory enhancement. Recently, memory reconsolidation was suggested to reinforce or integrate new information into reactivated memory. Here, we show that reactivated inhibitory avoidance (IA) memory is enhanced through reconsolidation under conditions in which memory extinction is not induced. This memory enhancement is mediated by neurons in the amygdala, hippocampus, and medial prefrontal cortex (mPFC) through the simultaneous activation of calcineurin-induced proteasome-dependent protein degradation and cAMP responsive element binding protein-mediated gene expression. Interestingly, the amygdala is required for memory reconsolidation and enhancement, whereas the hippocampus and mPFC are required for only memory enhancement. Furthermore, memory enhancement triggered by retrieval utilizes distinct mechanisms to strengthen IA memory by additional learning that depends only on the amygdala. Our findings indicate that reconsolidation functions to strengthen the original memory and show the dynamic nature of reactivated memory through protein degradation and gene expression in multiple brain regions.

**\*For correspondence:** kida@ nodai.ac.jp

[†]These authors contributed equally to this work

**Competing interests:** The authors declare that no competing interests exist.

**Reviewing editor**: Nahum Sonenberg, McGill University, Canada

## Introduction

Memory retrieval is not a passive phenomenon. Previous studies have presented evidence that memory retrieval is a dynamic process during which memories can be made stronger, weaker, or their content altered (*Misanin et al., 1968*; *Schneider and Sherman, 1968*; *Lewis, 1979*; *Mactutus et al., 1979*; *Gordon, 1981*; *Nader et al., 2000*; *Nader and Hardt, 2009*; *Dudai, 2012*). Recent studies have shown that reactivated memory becomes labile after retrieval and is re-stabilized through a gene expression-dependent process known as memory reconsolidation. Memory reconsolidation after retrieval may be used to maintain or update long-term memories, reinforcing or integrating new information into them (*Nader et al., 2000*; *Dudai, 2002*; *Tronel et al., 2005*). However, the function of memory reconsolidation still remains unclear; especially, whether memory reconsolidation strengthens the original memory (*Tronson et al., 2006*; *Inda et al., 2011*; *Pedroso et al., 2013*). Importantly, the reinforcement of traumatic memory after retrieval (i.e., re-experience such as flashbacks or nightmares) may be associated with the development of emotional disorders such as post-traumatic stress disorder (PTSD).

In classical Pavlovian fear conditioning paradigms, the reactivation of conditioned fear memory by re-exposure to the conditioned stimulus (CS) in the absence of the unconditioned stimulus (US) also initiates extinction as a form of new learning that weakens fear memory expression (i.e., a new CS-no US inhibitory memory that competes with the original CS-US memory trace) (*Pavlov, 1927*; *Rescorla, 2001*; *Myers and Davis, 2002*). Therefore, in the majority of reconsolidation paradigms, reactivation also includes extinction learning, which could confound how reconsolidation functions; the dominance

**eLife digest** Video cameras allow us to record events as they happen. When we look back at a video clip, what we see is an exact replica of what was originally recorded. We tend to assume that our memories work in a similar manner. However, recent research suggests that our memories may be more malleable than we realize. Once a memory has been reactivated, it goes through a process known as reconsolidation that can make it stronger or weaker, or that can change its content.

Now, Fukushima et al. have carried out a series of experiments which shed light on the process of memory reconsolidation. Mice were trained to remember a negative event, and later tested on their memory of this event. Some of the mice were also given a 'reactivation' session, during which they were reminded of the original memory. These mice were more fearful of the event during the memory test than those who had not been reminded of it. This suggests that the process of reconsolidating the memory after it had been retrieved had the effect of making the memory stronger.

Fukushima et al. then demonstrated that this enhancement depended on the synthesis of proteins in particular regions of the brain. When the mice were given an injection to block protein synthesis immediately after reactivation of the memory, their memory of the negative event was weakened. Crucially, this effect only happened when the injection was given immediately after reactivation of the memory; if the memory had not been reactivated, the injection did not change its strength.

Fukushima et al. went on to show that three regions of the brain—the amygdala, the hippocampus, and the medial prefrontal cortex—are involved in memory enhancement. However, only one of them, the amygdala, is involved in the other aspects of reconsolidation. This research could support clinical work by elucidating the potential role of reconsolidation in conditions such as post-traumatic stress disorder.

of the original or new memory traces is thought to determine the fate of memory through their competition (*Eisenberg et al., 2003*; *Pedreira and Maldonado, 2003*; *Suzuki et al., 2004*). Therefore, it is necessary to investigate the function of reconsolidation in an experimental condition in which memory extinction is not induced following memory retrieval.

To this end, we developed a procedure using an inhibitory avoidance (IA) task that can engage reconsolidation in the absence of extinction. In the IA task, mice receive an electrical footshock after they enter a dark (shock) compartment from a light (safe) compartment and form a memory to avoid the dark compartment. Re-exposure to the light compartment could elicit fear memory without giving the mice the opportunity to acquire extinction of the dark compartment memory. This is because the mice can only learn that the dark compartment does not lead to shock only after they have experienced this event by re-entering the dark compartment. This procedural control allows this paradigm to discriminate the reconsolidation and extinction phases at the time point when the mice enter the dark compartment from the light compartment. In the present study, we established a mouse model in which fear memory is enhanced after retrieval using the IA task and investigated the mechanisms of fear memory enhancement through reconsolidation.

## Results

### Enhancement of fear memory after retrieval in the IA task

The mice were first placed in the light compartment. A brief electrical footshock was delivered (Training) 5 s after they entered the dark compartment. The mice were re-exposed to the light compartment 24 hr later and we assessed their crossover latency to enter the dark compartment (Reactivation session). Immediately after they entered the dark compartment, the mice were returned to their home cage and crossover latency was assessed 48 hr later (post-reactivation long-term memory test, PR-LTM). The control group (treated with vehicle, VEH) displayed significantly increased crossover latency in Reactivation (398.9 ± 48.31 s) compared to Training (19.7 ± 1.68 s), indicating that the mice formed and retrieved IA memory (*Figure 1A*). Interestingly, the VEH group displayed further and significantly increased crossover latency at PR-LTM (1553.5 ± 193.81 s) compared to Reactivation,

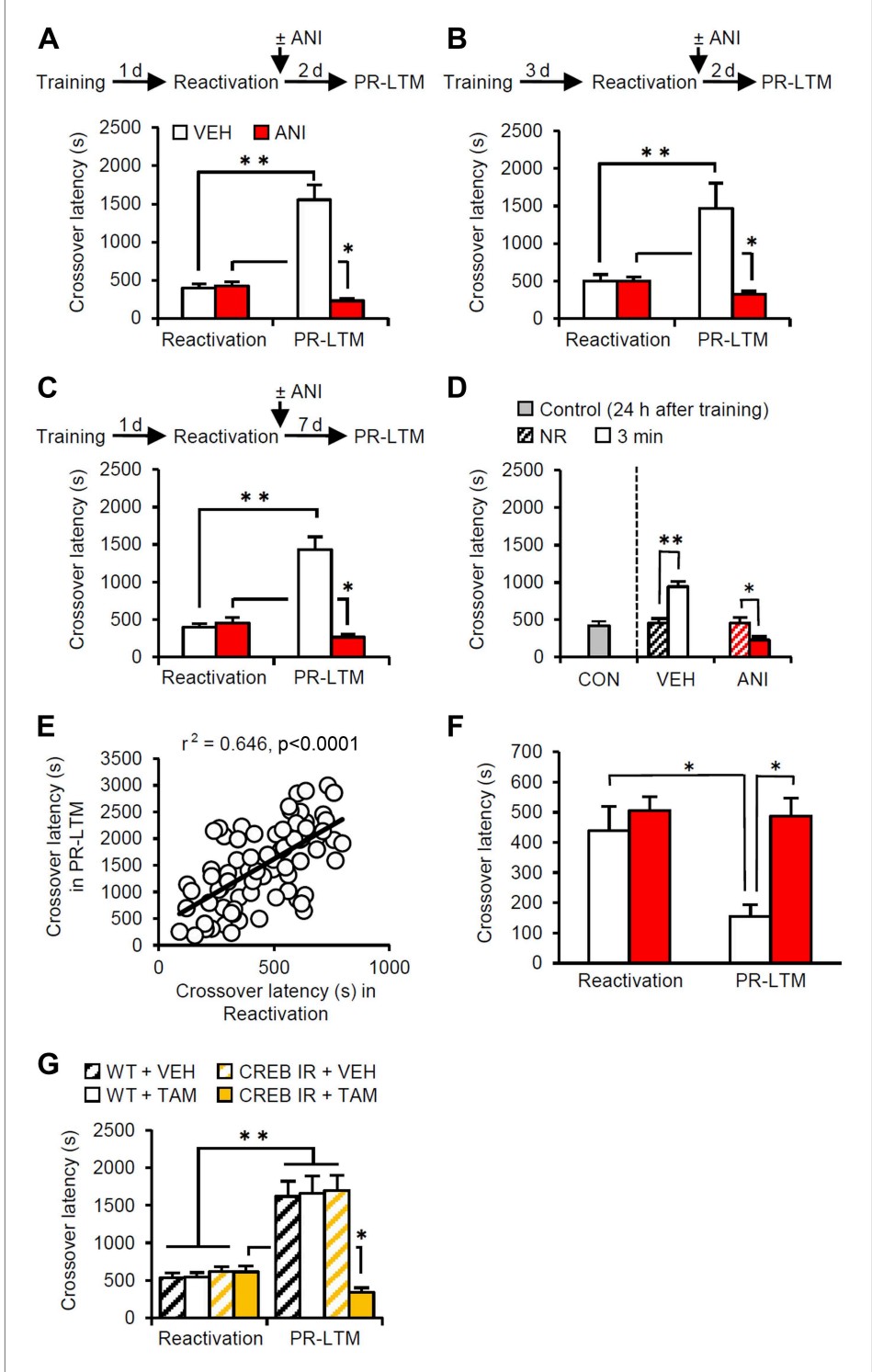

**Figure 1**. Memory retrieval can enhance inhibitory avoidance memory in a manner that is blocked by inhibiting protein synthesis. (**A**) Re-exposure to the light compartment until mice entered the dark compartment at 1 d after training. The VEH group showed enhancement of inhibitory avoidance (IA) memory (n = 10). The ANI group showed disruption of reactivated IA memory (n = 9). (**B**) At 3 d after training, a similar pattern was observed (VEH, n = 8; ANI, n = 12). (**C**) At 7 d after Reactivation (VEH, n = 8; ANI, n = 8). (**D**) Re-exposure to the light compartment for 3 min, but not 0 min (NR), led to IA memory enhancement (0 min: VEH, n = 8, ANI, n = 8; 3 min: VEH, n = 8, ANI, n = 9). (**E**) Positive correlation of crossover latency between the Reactivation and PR-LTM sessions
*Figure 1. Continued on next page*

*Figure 1. Continued*

(n = 96, $r^2$ = 0.646). (**F**) Re-exposure to the dark compartment for 3 min following re-exposure to the light compartment. The VEH group showed long-term extinction of IA memory, while ANI blocked this (VEH, n = 10; ANI, n = 10). (**G**) CREB-mediated transcription is required for memory reconsolidation in the protocol used in *Figure 1A* (WT/VEH, n = 9; WT/TAM, n = 9; CREB$^{IR}$/VEH, n = 9; CREB$^{IR}$/TAM, n = 9). ANI: anisomycin; CREB: cAMP responsive element binding protein; CREB$^{IR}$: inducible CREB repressor (CREB$^{IR}$) transgenic mice; IA: inhibitory avoidance; NR: non-reactivated; PR-LTM: post-reactivation long-term memory test; TAM: tamoxifen; VEH: vehicle; WT: wild-type mice. Error bars, SEM. *p<0.05, **p<0.005; paired *t* test. The results of the statistical analyses are presented in *Figure 1—source data 1*.

The following source data and figure supplements are available for figure 1:

**Source data 1**. Summary of statistical analyses with F values.

**Figure supplement 1**. Correlational analyses of crossover latency between the Training and Reactivation sessions.

suggesting that memory retrieval enhanced IA memory in our experimental condition. This ability of retrieval to enhance memories is consistent with previous work (*Gordon, 1981*).

Previous studies have shown that the inhibition of protein synthesis immediately after re-exposure to the light compartment disrupts reconsolidation of IA memory (*Milekic and Alberini, 2002*; *Milekic et al., 2007*). To this end, the mice received a systemic injection of anisomycin (ANI) immediately after Reactivation (*Figure 1A*). The ANI group showed decreased crossover latency at PR-LTM compared to Reactivation, although the VEH and ANI groups showed comparable crossover latency at Reactivation, suggesting that the inhibition of protein synthesis disrupted IA memory. These results suggest that memory retrieval induces reconsolidation, raising the intriguing hypothesis that memory enhancement may be mediated by reconsolidation.

To test whether this enhancement was specific only to these experimental parameters used in the previous experiment, we performed Reactivation or PR-LTM at 3 or 7 d after Training or Reactivation, respectively (*Figure 1B,C*). Consistent with the results shown in *Figure 1A*, the VEH and ANI groups displayed enhancement and disruption, respectively, of the reactivated memory at PR-LTM, although both groups displayed comparable crossover latency at Reactivation compared to the VEH and ANI groups, respectively, in *Figure 1A* (*Figure 1B* or *Figure 1C* vs *Figure 1A*). These results indicate that the enhancement associated with the reconsolidation of fear memory was not transient or specific to the age of the memory. We also examined the effects of a shorter duration of re-exposure to the light compartment on IA memory (*Figure 1D*). Trained mice were either not exposed to the light compartment (non-reactivated, NR) or were re-exposed to the light compartment for 3 min. The VEH and ANI groups in the NR condition displayed comparable crossover latency at PR-LTM (VEH, 457.25 ± 55.25 s; ANI, 450.38 ± 78.26 s). In contrast, re-exposure of the VEH and ANI groups to the light compartment for 3 min resulted in the enhancement and disruption, respectively, of the reactivated memory compared to the controls (0 min (no) re-exposure groups in Reactivation and control group re-exposed 24 hr after training, respectively). Taken together, our observations suggest that memory enhancement is associated with memory reconsolidation.

To clarify further the effects of Reactivation on the enhancement of IA memory, the crossover latency of individual mice from all VEH groups used in the present study (n = 96) was compared between Reactivation and PR-LTM. We observed a significant positive correlation of crossover latency between Reactivation and PR-LTM (*Figure 1E*). These observations indicate that longer reactivation of IA memory results in increased enhancement of this memory and strongly support our conclusion that IA memory is enhanced after retrieval. It is important to note that no correlation of crossover latency was observed between Training and Reactivation (*Figure 1—figure supplement 1*), indicating that enhancement of IA memory is associated with memory retrieval of this memory.

Finally, we examined whether our experimental paradigm dissociates the reconsolidation and extinction phases. To do this, the mice stayed in the dark compartment for 3 min after they entered from the light compartment at Reactivation (*Figure 1F*). The VEH and ANI groups showed decreased and comparable, respectively, crossover latency compared to Reactivation. These observations indicate that memory reactivation in the dark compartment induces long-term memory extinction that requires new gene expression. Thus, consistent with our hypothesis, the IA task enables us to discriminate

between the reconsolidation and extinction phases at the time point when the mice enter the dark compartment from the light compartment.

Previous studies indicated that cAMP responsive element binding protein (CREB)-mediated transcription is required for the reconsolidation of reactivated fear memory (*Kida et al., 2002*; *Mamiya et al., 2009*). Using inducible CREB repressor (CREB[IR]) transgenic mice (*Kida et al., 2002*), we tested what effect inhibition of CREB-mediated transcription would have in the protocol described in *Figure 1A* (*Figure 1G*). CREB[IR] mice and wild-type (WT) littermates received a systemic injection of tamoxifen (TAM) or VEH to inhibit CREB activity at 6 hr before Reactivation (*Kida et al., 2002*). The CREB[IR] mice injected with TAM and the other control groups displayed disrupted and enhanced, respectively, IA memory at PR-LTM. These observations suggest that CREB-mediated transcription is required for the reconsolidation/enhancement of IA memory.

## Identifying the brain systems activated by IA memory reconsolidation and enhancement

To characterize the brain systems that contribute to these memory representations, we tried to identify the brain regions in which CREB-mediated gene expression is activated. We measured the expression levels of the immediate-early genes c-fos and Arc, which are CREB-dependent genes, using immunohistochemistry (*Sheng et al., 1990*; *Kaczmarek and Robertson, 2002*; *Kawashima et al., 2009*). The following four groups were used in this experiment: two groups which were trained with a footshock (Trained [T] groups) and the remaining two groups which did not receive a footshock (Untrained [U] groups). During Reactivation, the animals were either returned to the light compartment (T/R and U/R) or not (T/NR and U/NR). Significant increases in the number of c-fos-positive cells were observed in the lateral (LA) and basolateral amygdala (BA), CA1 and CA3 regions of the hippocampus, and prelimbic (PL) and infralimbic (IL) regions of the medial prefrontal cortex (mPFC), but not in the central amygdala (CeA) regions or hippocampal dentate gyrus (DG) area, in the T/R group compared to the other control groups (*Figure 2A–D*). In contrast, there was no increase in c-fos expression in the anterior cingulate cortex (ACC), visual cortex (VC), temporal cortex (TC), perirhinal cortex (PRh), or entorhinal cortex (EC) regions of the T/R group (*Figure 2—figure supplement 1*). Similarly, significant increases in Arc expression were observed in the amygdala (LA and BA), hippocampus (CA1 and CA3), and mPFC (PL and IL) in the T/R group compared to the other control groups (*Figure 2—figure supplement 2*).

To understand further the functional roles of induced gene expression in the reconsolidation/enhancement of IA memory, we performed correlational analyses on the number of c-fos-positive cells and/or crossover latency among the brain regions of individual mice in the T/R condition. Interestingly, significant positive correlations of the number of c-fos positive cells were observed between the amygdala (LA and BA), hippocampus (CA1 and CA3), and mPFC (PL and IL) (*Table 1*), suggesting that c-fos expression in these regions was activated synchronously in response to memory reactivation. Significant positive correlations were also observed between crossover latency and the number of c-fos-positive cells (*Table 2*). Taken together with the finding shown in *Figure 1E*, our observations suggest that the increase in crossover latency at PR-LTM compared to Reactivation is associated with the synchronized induction of c-fos in the amygdala, hippocampus, and mPFC.

## Roles of gene expression in the amygdala, hippocampus, and mPFC in the reconsolidation/enhancement of IA memory

To examine the roles of new gene expression in the amygdala, hippocampus, and mPFC in the reconsolidation/enhancement of IA memory, we examined the effects of protein synthesis inhibition in these brain regions. We performed a similar experiment as in *Figure 1A*, except that the mice received a micro-infusion of ANI or VEH into the amygdala, hippocampus, or mPFC immediately after Reactivation. Similarly with the results for the systemic injection of ANI (*Figure 1A*), inhibition of protein synthesis in the amygdala resulted in the disruption of the reactivated IA memory (*Figure 2E*). Interestingly, protein synthesis inhibition in the hippocampus or mPFC failed to disrupt the reactivated IA memory, but blocked its enhancement (*Figure 2F,G*); the ANI groups displayed comparable crossover latency between Reactivation and PR-LTM. It is important to note that protein synthesis inhibition in both the hippocampus and mPFC blocked the enhancement of IA memory, but did not disrupt IA memory; furthermore, mice infused with ANI to the hippocampus or mPFC displayed a comparable enhancement of IA memory at PR-LTM-2 at 48 hr after PR-LTM-1 compared to the VEH group at PR-LTM-1 (*Figure 2— figure supplement 3*). Thus, these observations suggest that the inhibition of protein synthesis in the

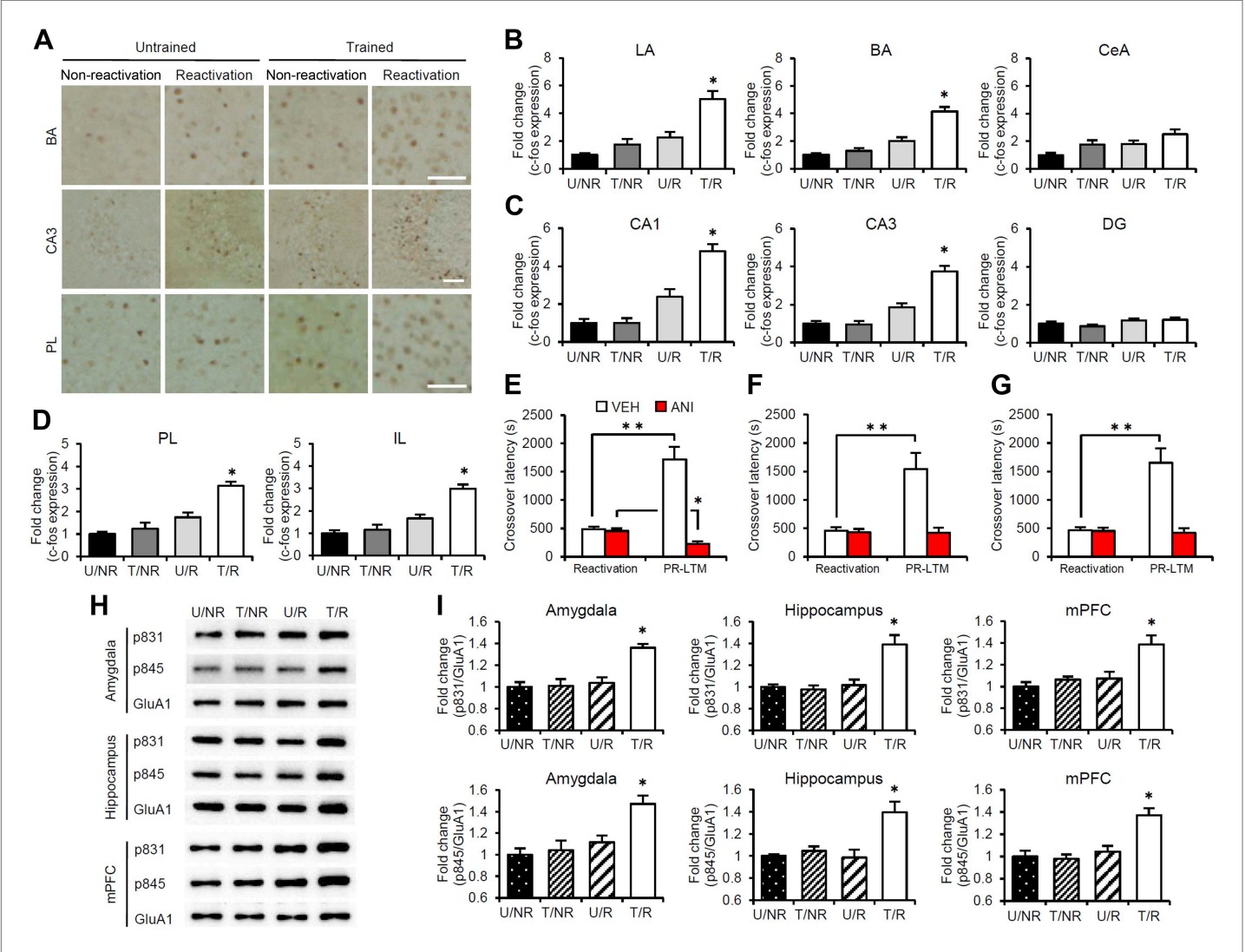

**Figure 2**. Roles of gene expression in the amygdala, hippocampus, and mPFC in the enhancement of reactivated inhibitory avoidance memory. (A–D) c-fos induction when IA memory is enhanced after reactivation. (A) Representative immunohistochemical staining of BA, CA3, and PL c-fos-positive cells from the indicated group. Scale bar, 50 μm. Two groups were trained with a footshock: one group received memory reactivation (T/R), while the other group did not (T/NR). The remaining two groups did not receive a footshock. During the Reactivation session, the animals were returned to the light compartment (U/R) or not (U/NR). (B–D) c-fos expression in the LA, BA, and CeA regions of the amygdala (B), CA1, CA3, and DG regions of the hippocampus (C), and PL and IL of the mPFC (D) (n = 13–21 for each group). (E–G) Effects of anisomycin micro-infusions immediately after Reactivation in the amygdala (E, VEH, n = 8, ANI, n = 9), hippocampus (F, VEH, n = 10, ANI, n = 10), and mPFC (G, VEH, n = 10, ANI, n = 11). Micro-infusion of ANI into the amygdala blocked IA memory as seen by the reduction in performance between Reactivation and PR-LTM. In contrast, micro-infusion of ANI into the hippocampus or mPFC blocked the enhancement, but not the underlying performance. (H and I) Phosphorylation of GluA1 at Ser831 and Ser845 was induced in the amygdala, hippocampus, and mPFC following memory retrieval. (H) Representative western blot analysis of the amygdala, hippocampus, and mPFC showing phosphorylated GluA1 and total GluA1 levels. (I) Levels of Ser831 and Ser845-phosphorylated GluA1 in the amygdala, hippocampus, and mPFC (n = 9 per group). The levels of Ser831- and Ser845-phosphorylated GluA1 for each group are expressed as the ratio of the U/NR group to the other groups. ANI: anisomycin; BA: basolateral amygdala; CeA: central amygdala; DG: dentate gyrus; IA: inhibitory avoidance; IL: infralimbic region; LA: lateral amygdala; mPFC: medial prefrontal cortex; PL: prelimbic region; PR-LTM: post-reactivation long-term memory test; VEH: vehicle. Error bars, SEM. *p<0.05, compared with the other control groups (B–D and I). **p<0.005; paired *t* test (E–G). The results of the statistical analyses are presented in *Figure 2—source data 1*.

The following source data and figure supplements are available for figure 2:

**Source data 1**. Summary of statistical analyses with F values (including the data for the figure supplements).

*Figure 2. Continued on next page*

*Figure 2. Continued*

**Figure supplement 1**. No c-fos induction in the ACC, VC, TC, PRh, and EC regions of the T/R group when inhibitory avoidance memory is enhanced after Reactivation.

**Figure supplement 2**. Arc induction when inhibitory avoidance memory is enhanced after Reactivation.

**Figure supplement 3**. Effects of inhibiting protein synthesis in the hippocampus and mPFC on the enhancement of inhibitory avoidance memory.

**Figure supplement 4**. Time course analysis of the phosphorylation levels of GluA1 at Ser831 and Ser845 following re-exposure to the light compartment.

hippocampus or mPFC at Reactivation simply blocks the enhancement of IA memory without modulating it. These results are consistent with a previous finding that protein synthesis in the amygdala, but not the hippocampus, is required for the reconsolidation of IA memory (*Taubenfeld et al., 2001*; *Milekic et al., 2007*), but more interestingly, suggest distinct roles for the amygdala and hippocampus/mPFC in the enhancement and reconsolidation of IA memory; new gene expression in the amygdala is required for the reconsolidation and enhancement of IA memory, while new gene expression in the hippocampus and mPFC is required only for its enhancement.

## Phosphorylation of GluA1 at Ser831 and Ser845 is induced in the mPFC, hippocampus, and amygdala regions when IA memory is enhanced

The α-amino-3-hydroxy-5-methyl-4-isoxazolepropionic acid receptor (AMPAR) GluA1 subunit undergoes distinct phosphorylation and dephosphorylation following the induction of long-term potentiation (LTP) and long-term depression (LTD) (*Lee et al., 2000*); serine 831 (Ser831) and serine 845 (Ser845) are phosphorylated by LTP induction or dephosphorylated after LTD induction, respectively. This phosphorylation is thought to alter the function of AMPAR and contribute to the expression of LTP and LTD. To examine the possibility that synaptic plasticity was changed following Reactivation, the phosphorylation of Ser831 and Ser845 in the synaptic membrane fraction of the amygdala, dorsal hippocampus, and mPFC regions was measured at 30 min after Reactivation. Significant increases in Ser831 and Ser845 phosphorylation were observed in the amygdala, hippocampus, and mPFC regions

**Table 1.** Significant positive correlations of the number of c-fos positive cells among amygdala (LA and BA), hippocampus (CA1 and CA3) and mPFC (PL and IL) (n = 18)

|  | PL | IL | ACC | CA1 | CA3 | DG | LA | BA | CeA | VC | TC | PRh | EC |
|---|---|---|---|---|---|---|---|---|---|---|---|---|---|
| PL |  | 0.831* | 0.520* | 0.639* | 0.576* | 0.624* | 0.640* | 0.650* | 0.639* | −0.123 | 0.205 | −0.134 | 0.015 |
| IL | 0.831* |  | 0.578* | 0.649* | 0.685* | 0.703* | 0.674* | 0.749* | 0.654* | 0.156 | 0.478 | −0.044 | 0.105 |
| ACC | 0.520* | 0.578* |  | 0.561* | 0.422* | 0.323* | 0.615* | 0.476* | 0.765* | −0.087 | 0.557 | 0.263 | 0.227 |
| CA1 | 0.639* | 0.649* | 0.561* |  | 0.530* | 0.401* | 0.676* | 0.629* | 0.577* | 0.154 | 0.536 | −0.114 | 0.145 |
| CA3 | 0.576* | 0.685* | 0.422* | 0.530* |  | 0.589* | 0.426* | 0.457* | 0.557* | −0.168 | 0.720 | 0.194 | 0.391 |
| DG | 0.624* | 0.703* | 0.323* | 0.401* | 0.589* |  | 0.395* | 0.326* | 0.517* | −0.210 | 0.336 | 0.075 | −0.165 |
| LA | 0.640* | 0.674* | 0.615* | 0.676* | 0.426* | 0.395* |  | 0.919* | 0.820* | −0.156 | 0.294 | 0.000 | 0.090 |
| BA | 0.650* | 0.749* | 0.476* | 0.629* | 0.457* | 0.326* | 0.919* |  | 0.768* | 0.074 | 0.074 | −0.118 | 0.164 |
| CeA | 0.639* | 0.654* | 0.765* | 0.577* | 0.557* | 0.517* | 0.820* | 0.768* |  | −0.187 | 0.420 | 0.100 | 0.207 |
| VC | −0.123 | 0.156 | −0.087 | 0.154 | −0.168 | −0.210 | −0.156 | 0.074 | −0.187 |  | 0.073 | −0.220 | 0.063 |
| TC | 0.205 | 0.478 | 0.557 | 0.536 | 0.720 | 0.336 | 0.294 | 0.296 | 0.420 | 0.073 |  | 0.583 | 0.748 |
| PRh | −0.134 | −0.044 | 0.263 | −0.114 | 0.194 | 0.075 | 0.000 | −0.118 | 0.100 | −0.220 | 0.583 |  | 0.579 |
| EC | 0.015 | 0.105 | 0.227 | 0.145 | 0.391 | −0.165 | 0.090 | 0.164 | 0.207 | 0.063 | 0.748 | 0.579 |  |

*indicate a significant positive correlation (p<0.05).

**Table 2.** Significant positive correlations between crossover latency and the number of c-fos-positive cells after the Reactivation session (n = 18)

| Region | PL | IL | ACC | LA | BA | CeA | CA1 | CA3 | DG | VC | TC | PRh | EC |
|---|---|---|---|---|---|---|---|---|---|---|---|---|---|
| Correlation coefficient | **0.574*** | **0.627*** | 0.202 | **0.612*** | **0.630*** | 0.346 | **0.646*** | **0.503*** | 0.360 | 0.571 | 0.362 | 0.074 | 0.202 |

*Significant positive correlation.
ACC: anterior cingulate cortex; BA: basolateral amygdala; CeA: central amygdala; DG: dentate gyrus; EC: entorhinal cortex; IL: infralimbic region; LA: lateral amygdala; PL: prelimbic region; PRh: perirhinal cortex; TC: temporal cortex; VC: visual cortex.

of the T/R groups compared to the control groups (**Figure 2H,I**). Furthermore, time course analyses indicated that increases in Ser831 and Ser845 phosphorylation in the T/R groups peaked at 30 min and returned to basal levels by 180 min after Reactivation (**Figure 2—figure supplement 4**). Our observations that Ser831 and Ser845 phosphorylation was induced in the amygdala, hippocampus, and mPFC following Reactivation suggest that synaptic plasticity was changed in these regions when IA memory was reconsolidated/enhanced.

### Neurons inducing c-fos activate proteasome-dependent protein degradation that is required for reconsolidation/enhancement

A previous study showed that proteasome-dependent protein degradation plays critical roles in the destabilization of reactivated contextual fear memory (**Lee et al., 2008**). Therefore, we investigated the roles of this protein degradation process in the brain systems regulating reconsolidation/enhancement. To do this, we compared c-fos induction and Lys48-linked poly-ubiquitin chain (Ub-Lys48), an activation marker of proteasome-dependent protein degradation, in the amygdala, hippocampus, and mPFC using immunohistochemistry. Consistent with the results shown in **Figure 2**, we observed increases in the number of c-fos-positive cells in the LA and BA of the amygdala, CA1 and CA3 of the hippocampus, and PL and IL of the mPFC only in the T/R group (**Figure 3A–F**). Interestingly, significant increases in the number of Ub-Lys48-positive cells were observed in the same regions of the T/R group where c-fos was induced (**Figure 3A–F**). Most importantly, more than 70–80% of the c-fos-positive neurons of the T/R group were also Ub-Lys48-positive (**Figure 3D–F**). These observations indicate that gene expression and proteasome-dependent degradation were induced in the same neurons following IA memory retrieval, suggesting that IA memory is reconsolidated/enhanced through gene expression and protein degradation in the same neurons.

To understand the roles of proteasome-dependent protein degradation in the reconsolidation/enhancement of IA memory, we examined the effects of inhibiting proteasome-dependent protein degradation by a micro-infusion of clasto-lactacystin β-lactone (β-lac) into the amygdala, hippocampus, or mPFC (**Figure 3G–I**). Interestingly, micro-infusions of β-lac into the amygdala, mPFC, or hippocampus blocked the enhancement of IA memory, indicating that activating proteasome-dependent protein degradation in these brain regions is required for the enhancement of IA memory. Furthermore, a micro-infusion of β-lac with ANI into the amygdala prevented the disruption of IA memory by protein synthesis inhibition (**Figure 3G**), suggesting that protein degradation in the amygdala is required for the destabilization of reactivated IA memory. Taken together, our observations suggest that IA memory is reconsolidated/enhanced in the amygdala through protein degradation and synthesis, whereas protein synthesis and degradation in the hippocampus and mPFC are required for the enhancement of IA memory.

### Distinct mechanism for memory reconsolidation/enhancement through retrieval and additional training

A previous study using contextual fear conditioning showed that memory reconsolidation mediates the strengthening of memories following additional training (re-learning) via the activation of proteasome-dependent protein degradation and gene expression (**Lee, 2008**). To compare the mechanisms of memory enhancement through retrieval and additional training, we examined the mechanisms of IA memory enhancement by additional training in our experimental condition (**Figure 4**). We performed a similar experiment as in **Figure 1A**, except that the mice received a footshock at Reactivation (Training-2) at 5 s after they entered the dark compartment and then received micro-infusions of drugs immediately after Training-2 (**Figure 4A**). Crossover latency increased dramatically in the VEH group at PR-LTM

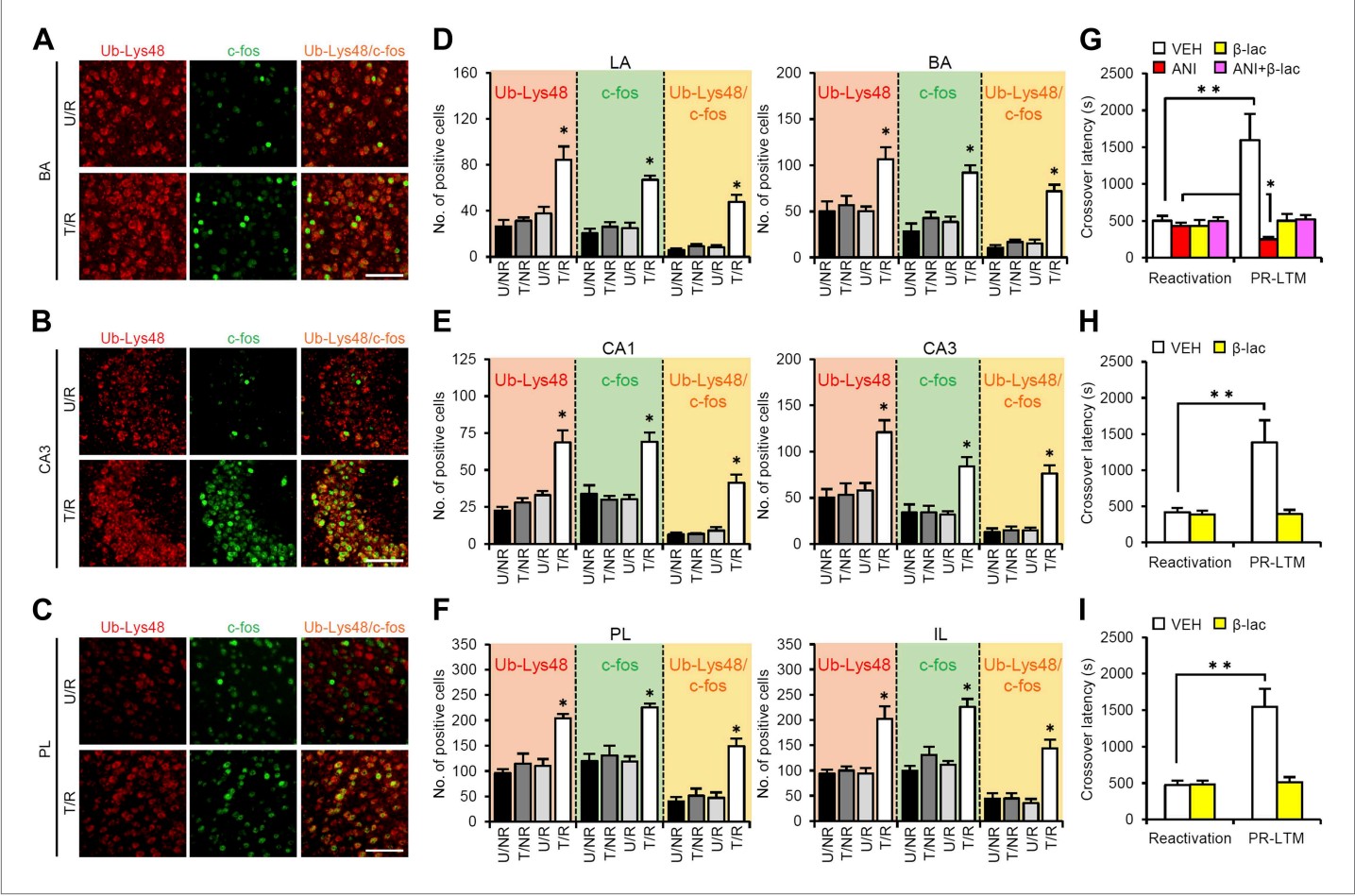

**Figure 3**. Roles of proteasome-dependent protein degradation in the amygdala, hippocampus, and mPFC in the enhancement of reactivated inhibitory avoidance memory. (**A–F**) Ub-Lys48 levels were increased following IA memory retrieval. (**A–C**) Representative immunohistochemical staining of BA (**A**), CA3 (**B**), and PL (**C**) Ub-Lys48-, c-fos-, and Ub-Lys48/c-fos-positive cells from the indicated mice. Scale bar, 100 μm. (**D–F**) Ub-Lys48, c-fos, and Ub-Lys48/c-fos expression in the LA and BA regions of the amygdala (**D**), CA1 and CA3 regions of the hippocampus (**E**), and PL and IL of the mPFC (**F**) (n = 5–6 for each group). (**G–I**) Effects of inhibition of proteasome-dependent protein degradation by micro-infusion of β-lac with or without ANI immediately after Reactivation into the amygdala (**G**), hippocampus (**H**), or mPFC (**I**) on the enhancement of IA memory (amygdala: VEH, n = 8, ANI, n = 8, β-lac, n = 8, ANI + β-lac, n = 9; hippocampus: VEH, n = 7, β-lac, n = 8; mPFC, VEH, n = 7, β-lac, n = 9). Error bars, SEM. *p<0.05, compared with the other control groups in Ub-Lys48-, c-fos-, or Ub-Lys48/c-fos-positive cells, respectively (**D–F**). ANI: anisomycin; β-lac, clasto-lactacystin β-lactone; BA: basolateral amygdala; IA: inhibitory avoidance; IL: infralimbic region; mPFC: medial prefrontal cortex; PL: prelimbic region; VEH: vehicle. *p<0.05, **p<0.005; paired *t* test (**G–I**). The results of the statistical analyses are presented in *Figure 3—source data 1*.

The following source data and figure supplements are available for figure 3:

**Source data 1**. Summary of statistical analyses with F values (including the data for the figure supplements).

**Figure supplement 1**. Roles of gene expression and proteasome-dependent protein degradation in the consolidation of inhibitory avoidance memory.

(3940 ± 571.5 s) at 48 hr after Training-2 compared to following Reactivation (without a footshock in *Figure 1A*), suggesting that additional training enhanced IA memory more than retrieval without a footshock. Similarly with a previous report, inhibition of protein synthesis in the amygdala disrupted IA memory, whereas inhibition of proteasome-dependent protein degradation with or without protein synthesis inhibition blocked the enhancement and disruption induced by ANI, respectively, of IA memory (*Figure 4B*). These results were consistent with previous findings that protein degradation and synthesis in the amygdala are required for strengthening fear memory mediated by additional training. More interestingly, in contrast to the results shown in *Figure 3*, the inhibition of protein synthesis in the hippocampus or mPFC did not affect the enhancement of IA memory by additional

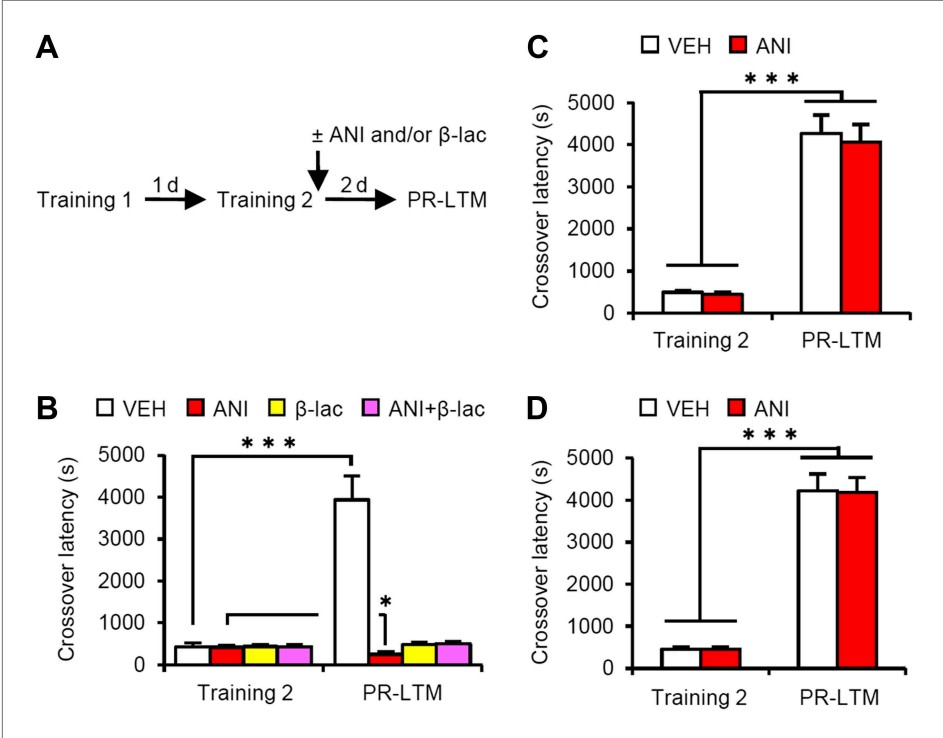

**Figure 4**. Effects of inhibiting protein synthesis and degradation in the amygdala, hippocampus, and mPFC on additional training. (**A**) Experimental design. (**B**) Effects of inhibiting protein synthesis and/or degradation immediately after Training-2 in the amygdala (VEH, n = 8, ANI, n = 8, β-lac, n = 8, ANI/β-lac, n = 9). (**C** and **D**) Effects of protein synthesis inhibition immediately after Training-2 in the hippocampus (**C**, VEH, n = 10, ANI, n = 10) or mPFC (**D**, VEH, n = 10, ANI, n = 10). The VEH group showed a dramatic enhancement of IA memory. Micro-infusion of ANI into the amygdala blocked IA memory as seen by the reduction in performance between Training-2 and PR-LTM. In contrast, micro-infusion of ANI into the hippocampus or mPFC did not affect additional training. ANI: anisomycin; β-lac, clasto-lactacystin β-lactone; IA: inhibitory avoidance; mPFC: medial prefrontal cortex; PR-LTM: post-reactivation long-term memory test; VEH: vehicle. Error bars, SEM. *p<0.05, ***p<0.001; paired *t* test. The results of the statistical analyses are presented in *Figure 4—source data 1*.

The following source data are available for figure 4:

**Source data 1**. Summary of statistical analyses with F values.

training (*Figure 4C,D*). Taken together, our observations that gene expression in the hippocampus and mPFC is required only for IA memory enhancement following retrieval indicate that these brain regions contribute to the enhancement of fear memory following retrieval, but not additional training, and more importantly, memory enhancement by retrieval utilizes a mechanism distinct from that used for additional learning.

Furthermore, it is important to note that: (1) protein synthesis inhibition in the hippocampus immediately after Training blocked the consolidation of IA memory (*Taubenfeld et al., 2001*; *Zhang et al., 2011*; *Figure 3—figure supplement 1*); (2) this consolidation was not affected by the inhibition of proteasome-dependent protein degradation in the hippocampus (*Figure 3—figure supplement 1*); and (3) an increase in c-fos (*Zhang et al., 2011*), but not Ub-Lys48, in the hippocampus was observed when IA memory was consolidated (*Figure 3—figure supplement 1*). Taken together, these observations indicate that the reconsolidation/enhancement observed in this study utilize a mechanism distinct from that used for additional learning and consolidation.

## Calcineurin is upstream of protein degradation
Previous studies have shown that molecules, such as LVGCCs and CB1, and protein degradation, which are required for memory extinction, also play critical roles in the destabilization of reactivated

memory (**Suzuki et al., 2008**; **Lee et al., 2008**). Interestingly, a recent study showed that calcineurin (CaN) activation in the hippocampus is required for the extinction of contextual fear memory (**de la Fuente et al., 2011**). Therefore, we examined the roles of CaN in the reconsolidation/enhancement of IA memory. We performed a similar experiment as in **Figure 3**, except that the mice received a micro-infusion of the CaN inhibitor FK506 at 5 min before Reactivation. Similar results were observed with those using the proteasome inhibitor β-lac (**Figure 5A–C**). The micro-infusion of FK506 into the amygdala blocked the disruption of reactivated IA memory by protein synthesis inhibition and enhancement of IA memory (**Figure 5A**), whereas the infusion of FK506 into the hippocampus or mPFC blocked the enhancement of IA memory (**Figure 5B,C**). These observations suggest that, similar to protein degradation, CaN activation in the amygdala is required for the destabilization and

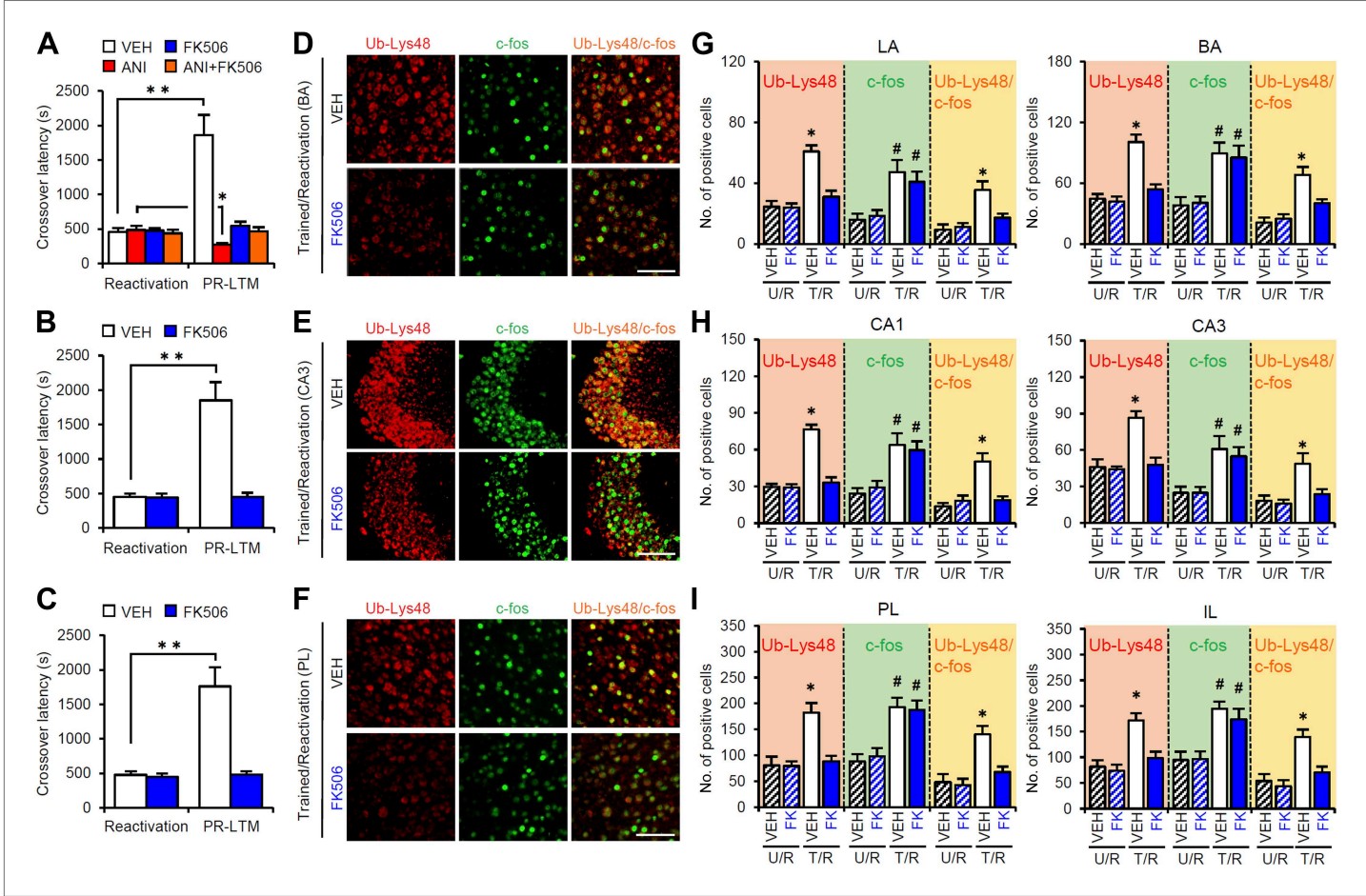

**Figure 5**. Roles of calcineurin in the amygdala, hippocampus, and mPFC on the enhancement of inhibitory avoidance memory and memory retrieval-induced protein degradation. (**A–C**) Effect of micro-infusion of FK506 with or without ANI before Reactivation into the amygdala (**A**), hippocampus (**B**), or mPFC (**C**) on the enhancement of IA memory (n = 10 for each group). (**D–I**) The calcineurin inhibitor FK506 blocked the increase of Ub-Lys48 following IA memory retrieval. (**D–F**) Representative immunohistochemical staining of BA (**D**), CA3 (**E**), and PL (**F**) Ub-Lys48-, c-fos-, and Ub-Lys48/c-fos-positive cells from the indicated mice. Scale bar, 100 μm. (**G–I**) Ub-Lys48, c-fos, and Ub-Lys48/c-fos expression in the LA and BA regions of the amygdala (**G**), CA1 and CA3 regions of the hippocampus (**H**), and PL and IL of the mPFC (**I**) (n = 8–11 for each group). ANI: anisomycin; BA: basolateral amygdala; IA: inhibitory avoidance; IL: infralimbic region; LA: lateral amygdala; mPFC: medial prefrontal cortex; PL: prelimbic region. Error bars, SEM. *p<0.05, **p<0.005; paired t test (**A–C**). *p<0.05, compared with the other control groups (**G–I**). #p<0.05, compared with the U/R VEH group. The results of the statistical analyses are presented in **Figure 5—source data 1**.

The following source data and figure supplements are available for figure 5:

**Source data 1**. Summary of statistical analyses with F values (including the data for the figure supplements).

**Figure supplement 1**. Roles of calcineurin on the enhancement of inhibitory avoidance memory after retrieval.

enhancement of IA memory, while CaN activation in the hippocampus and mPFC is required for the enhancement of IA memory.

To understand the roles of CaN in the activation of gene expression and proteasome-dependent protein degradation, we examined the effects of CaN inhibition on the number of c-fos- and Ub-Lys48-positive cells following Reactivation, similar to the experiment shown in *Figure 3*, except that the mice received a systemic injection of FK506 (5 mg/kg) or VEH at 5 min before Reactivation (*Figure 5D–I*). It is important to note that, similar to the results of the micro-infusions, the systemic injection of FK506 blocked the disruption of the reactivated IA memory by protein synthesis inhibition and the enhancement of IA memory (*Figure 5—figure supplement 1*). The T/R groups treated with or without FK506 showed a comparable increase in the number of c-fos-positive cells compared to the U/R groups (*Figure 5D–I*), suggesting that CaN activation is not required for c-fos induction. In contrast, the T/R groups treated with FK506 did not show an increase in the number of Ub-Lys48-positive cells; only the T/R groups treated with VEH showed significantly more Ub-Lys48-positive cells than the other groups (*Figure 5D–I*). Thus, the inhibition of CaN blocked the increase of Ub-Lys48-positive cells, but not c-fos-positive cells, in the amygdala, hippocampus, and mPFC. It is important to note that similar results were observed using another CaN inhibitor, cyclosporine A (*Figure 5—figure supplement 1*). These observations suggest that CaN functions as an upstream regulator of proteasome-dependent protein degradation, but not c-fos induction, when reactivated memory is reconsolidated/enhanced.

## Discussion

Retrieval has been thought to have a role in enhancing memory that requires activation of gene expression and signal transduction pathways such as mTOR (*Gordon, 1981*; *Nader et al., 2000*; *Nader and Hardt, 2009*; *Inda et al., 2011*; *Pedroso et al., 2013*). In this study, we developed IA paradigms in which the retrieval of fear memory triggers reconsolidation without inducing memory extinction; the reconsolidation and extinction phases are discriminated at the time point when the mice enter a dark compartment from a light compartment. Using this paradigm, we found that memory retrieval enhances IA memory through reconsolidation. In contrast to the mechanisms for the reinforcement of contextual fear memory by additional learning, the enhancement of IA memory by retrieval required CREB-mediated gene expression and calcineurin-induced proteasome-dependent protein degradation not only in the amygdala but also in the hippocampus and mPFC. Consistently, IA memory retrieval was suggested to induce synaptic plasticity in these brain regions through the phosphorylation of AMPAR. Interestingly, we further found that the amygdala is required for the reconsolidation and enhancement of IA memory, whereas the hippocampus and mPFC are required for the enhancement, but not reconsolidation, of IA memory. These findings suggest that an IA memory is enhanced/reconsolidated through the reactivation of memory circuits consisting of multiple brain regions including the amygdala, hippocampus, and mPFC and that the amygdala plays central and distinct roles from the hippocampus and mPFC in the enhancement/reconsolidation of IA memory. It is important to investigate further the differences in the roles of the hippocampus and mPFC in the enhancement of an IA memory.

A previous study showed that gene expression is required for the re-stabilization (reconsolidation) of reactivated memory, whereas proteasome-dependent protein degradation is required for its destabilization (*Nader et al., 2000*; *Taubenfeld et al., 2001*; *Debiec et al., 2002*; *Kida et al., 2002*, *Lee et al., 2008*). We found that the amygdaloid neurons reactivated by the retrieval of an IA memory activated proteasome-dependent protein degradation and gene expression, suggesting that these reactivated neurons regulate the destabilization and re-stabilization of IA memory. Importantly, our finding that inhibition of CaN blocked proteasome-dependent protein degradation, but not gene expression, suggests that destabilization and re-stabilization are regulated independently at the early stages of the signal transduction pathways activated after memory retrieval. Furthermore, our finding that the activation of gene expression and proteasome-dependent protein degradation in the hippocampus and mPFC is required for the enhancement, but not reconsolidation, of an IA memory also suggests that the activation of proteasome-dependent protein degradation is not only required for the destabilization of a reactivated memory but also plays additional roles in the modification or alteration of memory without the induction of destabilization.

Previous and current studies have shown that CB1, proteasome-dependent protein degradation, and CaN are required for not only the destabilization of reactivated fear memory but also the extinction of fear memory (*Suzuki et al., 2008*; *Lee et al., 2008*; *de la Fuente et al., 2011*). These findings

suggest that destabilization and extinction, both of which are triggered by memory retrieval, share similar signal transduction cascades. It is important to investigate further and compare the molecular mechanisms that underlie the destabilization and extinction of reactivated fear memory. Understanding such mechanisms will enable the identification of the mechanism by which the fate of a memory is determined, that is, reconsolidation or extinction.

## Materials and methods

### Mice

All experiments were conducted according to the Guide for the Care and Use of Laboratory Animals (Japan Neuroscience Society) and the Guide for the Tokyo University of Agriculture. All of the animal experiments performed in this study were approved by the Animal Care and Use Committee of Tokyo University of Agriculture (authorization number: 250013). Male C57BL/6N mice were obtained from Charles River (Yokohama, Japan). Transgenic mice expressing an inducible CREB repressor (CREB$^{IR}$ mice) were backcrossed to C57BL/6N mice (National Institutes of Health) (*Kida et al., 2002*; *Suzuki et al., 2008*; *Mamiya et al., 2009*). The mice were housed in cages of five or six animals each, maintained on a 12 hr light/dark cycle, and allowed ad libitum access to food and water. The mice were at least 8 weeks of age when tested. Testing was performed during the light phase of the cycle. All experiments were conducted blind to the treatment condition of the mice.

### IA test

The step-through IA apparatus (OHARA Pharmaceutical, Tokyo, Japan) consisted of a box with separate light and dark compartments (both 15.5 × 12.5 × 11.5 cm). The light compartment was illuminated by a fluorescent light (2500 lux) (*Fukushima et al., 2008*; *Zhang et al., 2011*). Before the commencement of IA, the mice were handled individually for 2 min each day for 1 week. During the training sessions, each mouse was allowed to habituate to the light compartment for 30 s, and the guillotine door was raised to allow access to the dark compartment. Latency to enter the dark compartment was considered as a measure of acquisition. As soon as the mice had entered the dark compartment, the guillotine door was closed. After 5 s, a footshock (0.2 mA) was delivered for a total period of 2 s (training). At 24 hr after the training session, the mice were placed back in the light compartment (Reactivation) for a varying length of time (0 min, 3 min, or until the mice entered the dark compartment without a footshock). Memory was assessed at 48 hr later (PR-LTM) as the crossover latency for the mice to enter the dark compartment when replaced in the light compartment, as in Reactivation.

For the first experiment, we examined the effect of protein synthesis inhibition after Reactivation (re-exposure to the light compartment). The protein synthesis inhibitor anisomycin (ANI; Wako, Osaka, Japan) was dissolved in saline (pH adjusted to 7.0–7.4 with NaOH). The mice were trained as described above, and at 24 hr or 3 d later they received VEH or ANI (150 mg/kg, i.p.) immediately after re-exposure to the light compartment for 0 min, 3 min, or until they entered the dark compartment without a footshock (Reactivation). At 48 hr or 7 d after the Reactivation session, the mice were once again placed in the light compartment, and crossover latency was assessed. At this dose, ANI inhibits >90% of protein synthesis in the brain during the first 2 hr (*Flood et al., 1973*).

To examine the effects of re-exposure to the dark compartment on memory extinction, the mice were re-exposed to the light compartment until they entered the dark compartment. Immediately after the mice had entered the dark compartment, the guillotine door was closed and the mice stayed in the dark compartment for 3 min without a footshock (Reactivation). The mice received VEH or ANI (150 mg/kg, i.p.) immediately after Reactivation. At 48 hr later, crossover latency was assessed again (PR-LTM).

To examine the effects of disrupting CREB function on memory reconsolidation, we used transgenic mice that express an inducible CREB repressor (CREB$^{IR}$) in the forebrain, where a dominant-negative CREB protein is fused with the ligand binding domain of a mutant estrogen receptor (ER). Previous studies have shown that the systemic injection of TAM, the artificial ligand for ER, into these transgenic mice inhibits CREB activity in the forebrain (*Kida et al., 2002*). At 24 hr after the training session, CREB$^{IR}$ and WT mice were placed back in the light compartment and then crossover latency was assessed (Reactivation). The mice were administered an intraperitoneal injection of 16 mg/kg 4-hydroxytamoxifen (TAM; Sigma, MO, USA), which was dissolved in 10 ml peanut oil (Sigma) or VEH (a similar volume of

peanut oil), at 6 hr before retrieval (*Kida et al., 2002*; *Suzuki et al., 2008*; *Mamiya et al., 2009*). At 48 hr after Reactivation, the mice were once again placed in the light compartment, and crossover latency was assessed (PR-LTM).

For the second experiment (c-fos, Arc, and Ub-Lys48 immunohistochemistry), we examined the brain regions that are activated after re-exposure to the light compartment. The mice were divided into four groups: (1) T/R and T/NR groups; two groups of mice were trained as described above, and at 24 hr later, were or were not re-exposed to the light compartment. The animals were then anesthetized with Nembutal (750 mg/kg, i.p.) at 90 min after Reactivation; (2) U/R and U/NR groups; two groups received a training session in the absence of footshock, and at 24 hr later, were or were not re-exposed to the light compartment. The animals were then anesthetized, as described above, at 90 min after Reactivation.

For the third experiment (micro-infusion of drugs), we examined the effects of the inhibition of protein synthesis, proteasome-dependent protein degradation, and CaN in the amygdala, hippocampus, or mPFC on memory reconsolidation/enhancement. ANI was dissolved in artificial cerebrospinal fluid (ACSF) and adjusted to pH 7.4 with NaOH. The proteasome inhibitor β-lac (Sigma) was dissolved in 2% dimethyl sulfoxide (DMSO) in 1 M HCl diluted in ACSF and adjusted to pH 7.0–7.4 with NaOH (*Lee et al., 2008*). The CaN inhibitor FK506 monohydrate (FK506; Sigma) was dissolved in ACSF containing three drops of Tween 80 in 2.5 ml of 7.5% DMSO and adjusted to pH 7.4 with NaOH. The mice were trained as described above, and at 24 hr later, they were placed back in the light compartment (Reactivation). The mice were micro-infused with ANI (62.5 µg), β-lac (9.6 ng), FK506 (10 µg), or VEH immediately into the various brain regions after (*Figures 2–4*) or 5 min before (*Figure 5*) Reactivation. At 48 hr after Reactivation, the mice were once again placed in the light compartment and crossover latency was assessed (PR-LTM). Micro-infusions into the hippocampus and mPFC (0.5 µl) were performed at a rate of 0.25 µl/min. Micro-infusions into the amygdala (0.2 µl) were performed at a rate of 0.1 µl/min. The injection cannula was left in place for 2 min after micro-infusion and the mice were then returned to their home cages.

For the fourth experiment (additional training), we compared the mechanisms underlying memory enhancement through retrieval and additional training. The mice received a footshock in the reactivation (Training-2) session at 5 s after they entered the dark compartment and then received micro-infusions of drugs immediately after Training-2. At 48 hr later, crossover latency was assessed (PR-LTM). The dose of locally infused ANI used inhibits 90% of protein synthesis for at least 4 hr (*Rosenblum et al., 1993*).

For the fifth experiment (treatment with the CaN inhibitor), we examined the effects of CaN inhibition on the expression of c-fos and Ub-Lys48 following Reactivation. The CaN inhibitor FK506 and cyclosporin A (Cyc; Wako) were dissolved in saline containing one drop of Tween 80 in 3 ml of 2.5% DMSO and 10% Cremophor EL (Sigma). The mice received a systemic injection of FK506 (5 mg/kg), Cyc (5 mg/kg), or VEH at 5 min before Reactivation (re-exposure to the light compartment). For immunohistochemistry, the animals were anesthetized at 90 min after Reactivation, as described above. Another group of mice was assessed for crossover latency at 48 hr after Reactivation (PR-LTM).

## Immunohistochemistry

Immunohistochemistry was performed as described previously (*Mamiya et al., 2009*; *Suzuki et al., 2011*; *Zhang et al., 2011*). After anesthetization, all mice were perfused with 4% paraformaldehyde. The brains were then removed, fixed overnight, transferred to 30% sucrose, and stored at 4°C. Coronal sections (30 µm) were cut in a cryostat.

For c-fos or Arc staining, the sections were washed and preincubated in 3% $H_2O_2$ in methanol for 1 hr, followed by incubation in a blocking solution (phosphate-buffered saline [PBS] plus 1% goat serum albumin, 1 mg/ml bovine serum albumin, and 0.05% Triton X-100) for 3 hr. Consecutive sections were incubated with a polyclonal rabbit primary antibody for anti-c-fos (Ab-5; 1:5000; Millipore, MA, USA) or anti-Arc (1:1000; Santa Cruz Biotechnology, TX, USA) in the blocking solution overnight. Subsequently, the sections were washed with PBS and incubated for 3 hr at room temperature with biotinylated goat anti-rabbit IgG (SAB-PO Kit; Nichirei Biosciences, Tokyo, Japan), followed by 1 hr at room temperature in the streptavidin-biotin-peroxidase complex (SAB-PO Kit).

For Ub-Lys48 and/or c-fos staining, free floating sections were treated with 1% $H_2O_2$ and then incubated overnight with the rabbit monoclonal anti-Ub-Lys48 antibody (1:100; Millipore) and/or

rabbit polyclonal anti-c-fos antibody (1:1000; Millipore) in the blocking solution as described above. The sections were washed with PBS and then incubated with horseradish peroxidase-conjugated donkey anti-rabbit IgG (1:500; Jackson ImmunoResearch Laboratories, PA, USA) for c-fos or biotinylated donkey anti-rabbit IgG (1:500; Jackson ImmunoResearch Laboratories) for Ub-Lys48 for 1 hr at room temperature. Ub-Lys48 signals were amplified and visualized using a VECTASTAIN Elite ABC Kit (Vector Laboratories, CA, USA) and Alexa Fluor-conjugated streptavidin (Invitrogen, OR, USA). c-fos signals were amplified with TSA-FCM (Invitrogen). The sections were mounted on slides and coverslipped using mounting medium (Millipore).

## Quantification

Structures were defined anatomically according to the atlas of *Franklin and Paxinos (1997)*. All immunoreactive neurons were counted by an experimenter blind to the treatment condition.

Quantification of c-fos- or Arc-positive cells in sections (100 × 100 µm) of the mPFC (bregma between 2.10 and 1.98 mm), amygdala (bregma between −1.22 and −1.34 mm), dorsal hippocampus (bregma between −1.46 and −1.82 mm), VC (bregma between −3.88 and −4.00), ACC (bregma between 0.8 and 1.0 mm), TC (bregma between −3.88 and −4.00 mm), PRh (bregma between −3.88 and −4.00 mm), and EC (bregma between −3.88 and −4.00 mm) was performed using a computerized image analysis system, as described previously (WinROOF version 5.6 software; Mitani Corporation, Fukui, Japan) (*Frankland et al., 2006*; *Suzuki et al., 2008*, *2011*; *Mamiya et al., 2009*; *Zhang et al., 2011*). Immunoreactive cells were counted bilaterally with a fixed sample window across at least three sections.

Fluorescence images were acquired using a confocal microscope FV300 (Olympus, Tokyo, Japan) or TCS SP8 (Leica, Wetzlar, Germany). For Ub-Lys48 and/or c-fos staining, confocal 2 µm z-stack images were obtained using LAS AF software (Leica). Equal cutoff thresholds were applied to all slices. We quantified the number of Ub-Lys48- or c-fos-positive cells using a 20× (for the mPFC and amygdala) or 40× (for the hippocampus) objective. The cells in the field of view within the mPFC and amygdala (581 × 581 µm) and the hippocampus (290 × 290 µm) across at least two sections were counted using WinROOF version 5.6 software, as described above.

## Surgery for drug micro-infusion

Surgery was performed as described previously (*Frankland et al., 2006*; *Suzuki et al., 2008*, *2011*; *Mamiya et al., 2009*; *Kim et al., 2011*; *Zhang et al., 2011*; *Nomoto et al., 2012*). Under Nembutal anesthesia and using standard stereotaxic procedures, a stainless steel guide cannula (22 gauge) was implanted into the mPFC, dorsal hippocampus, or amygdala. Stereotaxic coordinates for mPFC, dorsal hippocampus, or amygdala placement based on the brain atlas of *Franklin and Paxinos (1997)* were as follows: mPFC (2.7 mm, ±0 mm, −1.6 mm), dorsal hippocampus (−1.8 mm, ±1.8 mm, −1.9 mm), or amygdala (−1.3 mm, ±3.3 mm, −4.4 mm). The mice were allowed to recover for at least 1 week after surgery. After this, they were handled for 1 week before the commencement of IA. Only mice with a cannulation tip within the boundaries of the amygdala, hippocampus, or mPFC were included in the data analysis. Cannulation tip placement is shown in *Figure 6*.

## Tissue preparation and western blotting

Mouse brains were sliced using a Rodent Brain Matrix (RBM-2000; ASI Instruments, MI, USA). The amygdala (bregma between −1.06 and −2.06 mm), dorsal hippocampus (bregma between −1.06 and −2.06 mm), and mPFC (bregma between 2.4 and 1.4 mm) regions were punched with a Sample corer (0.8 mm inner diameter; Muromachi, Tokyo, Japan) and stored at −80°C. Synaptic membrane fractions were isolated on a discontinuous sucrose gradient, as described previously (*Van den Oever et al., 2008*; *Counotte et al., 2011*). Brain tissues were homogenized in ice-cold homogenization buffer (4 mM HEPES, pH 7.4; 320 mM sucrose) containing a protease inhibitor mixture (Complete; Roche Diagnostics, IN, USA). The homogenized samples were centrifuged twice at 500×$g$ at 4°C for 2 min to remove nuclei and other debris. The supernatant was centrifuged at 20,000×$g$ at 4°C for 30 min. The pellets were suspended in a sodium dodecyl sulfate–polyacrylamide gel loading buffer and analyzed by western blotting.

Western blotting, using a rabbit polyclonal anti-GluR1 antibody (1:1000; Millipore), anti-glutamate receptor 1 phospho-Ser831 antibody (1:1000; Millipore), or anti-glutamate receptor 1 phospho-Ser845 antibody (1:1000; Millipore), was performed as described previously (*Hosoda et al., 2004*). Positive antibody binding was visualized using an ImmunoStar LD system (Wako), and protein-transferred

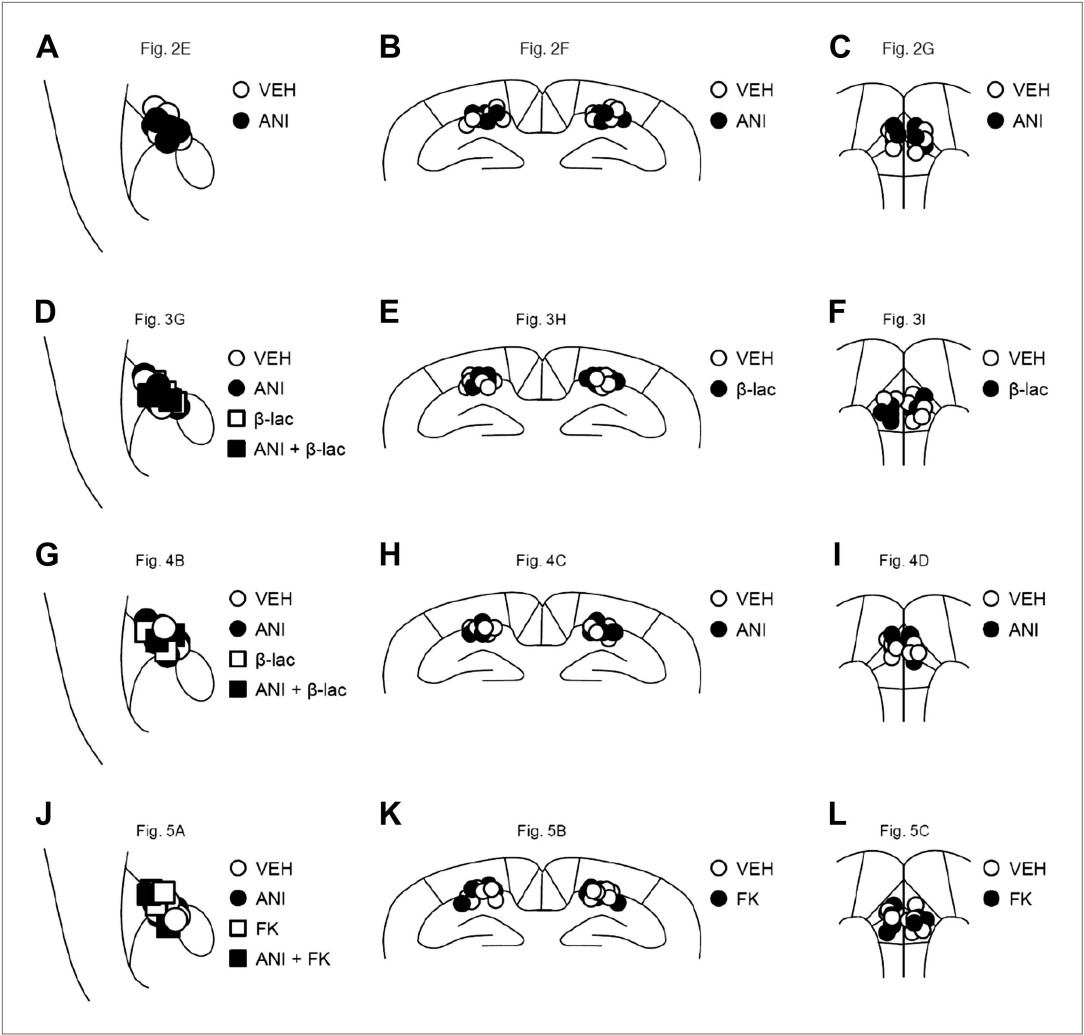

**Figure 6**. Cannula tip placement in the amygdala, hippocampus, and mPFC. (**A**–**L**) Cannula tip placement from mice infused with each drug shown in *Figure 2E* (**A**), *Figure 2F* (**B**), *Figure 2G* (**C**), *Figure 3G* (**D**), *Figure 3H* (**E**), *Figure 3I* (**F**), *Figure 4B* (**G**), *Figure 4C* (**H**), *Figure 4D* (**I**), *Figure 5A* (**J**), *Figure 5B* (**K**), and *Figure 5C* (**L**). Schematic drawing of coronal sections from all micro-infused animals (amygdala, 1.34 mm posterior to the bregma; hippocampus, 1.94 mm posterior to the bregma; mPFC, 1.94 mm anterior to the bregma). Only mice with needle tips within the boundaries of the amygdala, hippocampus, or mPFC were included in the data analysis. ANI: anisomycin; β-lac: clasto-lactacystin-β-lactone; FK: FK506; VEH: vehicle.

PVDF membranes (Bio-Rad, CA, USA) were analyzed using the Lumi-imager TM chemiluminescence detection system (Roche Diagnostics, IN, USA). The phosphorylation levels of GluA1 were calculated by normalizing the levels of phosphorylated GluA1 at Ser831 or Ser845 to the total amount of GluA1 (relative phospho-GluA1 [Ser831 or Ser845]/GluA1 levels).

## Data analysis

One-way analysis of variance (ANOVA) followed by post hoc Newman–Keuls and two- or three-way ANOVA followed by post hoc Bonferroni's comparisons were used to analyze the effects of group, genotypes, training, reactivation, and drugs. A repeated ANOVA followed by post hoc Bonferroni's comparisons were used to analyze the effects of drugs and times on crossover latency. A paired *t* test was used to analyze the differences in the crossover latency within each group between two sessions (Training vs Reactivation, Reactivation vs PR-LTM, or PR-LTM-1 vs PR-LTM-2). A two-tailed, paired Student's *t* test was used to analyze the GluA1 phosphorylation levels at each time point. All values in the text and figure legends are means ± standard error of the mean (SEM).

## Acknowledgements

SK was supported by Grant-in-Aids for Scientific Research (B) (23300120 and 20380078) and Challenging Exploratory Research (24650172), Grant-in-Aids for Scientific Research on Priority Areas -Molecular Brain Science- (18022038 and 22022039), Grant-in-Aid for Scientific Research on Innovative Areas (Research in a proposed research area) (24116008, 24116001 and 23115716), Core Research for Evolutional Science and Technology (CREST), Japan, The Sumitomo Foundation, Japan and the Takeda Science Foundation, Japan.

## Additional information

### Funding

| Funder | Grant reference number | Author |
| --- | --- | --- |
| Japan Society for the Promotion of Science (JSPS) | Grant-in-Aids for Scientific Research (B), Japan (23300120, 20380078) | Satoshi Kida |
| Japan Society for the Promotion of Science (JSPS) | Grant-in-Aids for Challenging Exploratory Research, Japan (24650172) | Satoshi Kida |
| Japan Society for the Promotion of Science (JSPS) | Grant-in-Aid for Scientific Research on Priority Areas, Molecular Brain Science (18022038, 22022039) | Satoshi Kida |
| Japan Society for the Promotion of Science (JSPS) | Grant-in-Aid for Scientific Research on Innovative Areas, Japan (24116008, 24116001, 23115716) | Satoshi Kida |
| Core Research for Evolutional Science and Technology, Japan Science and Technology Agency (CREST, JST) | | Satoshi Kida |
| The Sumitomo Foundation, Japan | | Satoshi Kida |
| The Takeda Science Foundation, Japan | | Satoshi Kida |

The funder had no role in study design, data collection and interpretation, or the decision to submit the work for publication.

### Author contributions

HF, Performed behavioral and western blot analyses, Acquisition of data, Analysis and interpretation of data; YZ, Performed immunohistochemical analyses, Acquisition of data, Analysis and interpretation of data; GA, RI, KN, Supervised experimental analyses, Analysis and interpretation of data; SK, Responsible for the hypothesis development and overall design of the research and experiment, and supervised the experimental analyses, Conception and design, Analysis and interpretation of data, Drafting or revising the article, Contributed unpublished essential data or reagents

### Ethics

Animal experimentation: All experiments were conducted according to the Guide for the Care and Use of Laboratory Animals (Japan Neuroscience Society) and the Guide for the Tokyo University of Agriculture. All the animal experiments were approved by the Animal Care and Use Committee of Tokyo University of Agriculture (authorization number: 250013). All surgical procedures were performed under Nembutal anesthesia, and every effort was made to minimize suffering.

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
