## [Decision Letter]

Thank you for sending your work entitled “Enhancement of fear memory by retrieval through reconsolidation” for consideration at *eLife*. Your article has been favorably evaluated by a Senior editor and 2 reviewers, one of whom is a member of our Board of Reviewing Editors.

The Reviewing editor and the other reviewer discussed their comments before we reached this decision. There are only minor points to be addressed before acceptance and the Reviewing editor has assembled the following comments to help you prepare a revised submission.

The study by Fukushima et al. demonstrates that inhibitory avoidance fear memory in mice is enhanced following retrieval via reconsolidation. Moreover, the authors show the dependence of the memory enhancement on CREB-dependent transcription and calcineurin-induced proteasome activation. The authors use various behavioural, biochemical, genetic and immunochistochemical techniques in the study. To our knowledge, this manuscript represents one of the most important contributions to memory enhancement through memory reconsolidation. It outlines the contributions of several brain regions (amygdala, hippocampus and medial prefrontal) and signaling systems (CREB transcription and CaN-dependent protein degradation) to this important process. Central to this impressive paper is the behavioral approach the authors developed for studies of memory reconsolidation.

However, the authors fail to mention previous studies showing that inhibitory avoidance memory in rats can be enhanced by non-reinforced retrieval (Pedroso et al. J Neural Transm (2013) 120:1525-1531, and Inda et al, The Journal of Neuroscience, February 2, 2011, 31(5):1635-1643). These studies also documented that the memory enhancement phenomenon requires protein synthesis and mTOR activity, and they should be cited and discussed in a resubmission.

---

## [Author Response]

*[…] To our knowledge, this manuscript represents one of the most important contributions to memory enhancement through memory reconsolidation. It outlines the contributions of several brain regions (amygdala, hippocampus and medial prefrontal) and signaling systems (CREB transcription and CaN-dependent protein degradation) to this important process. Central to this impressive paper is the behavioral approach the authors developed for studies of memory reconsolidation*.

*However, the authors fail to mention previous studies showing that inhibitory avoidance memory in rats can be enhanced by non-reinforced retrieval (Pedroso et al. J Neural Transm (2013) 120:1525-1531, and Inda et al, The Journal of Neuroscience, February 2, 2011, 31(5):1635-1643). These studies also documented that the memory enhancement phenomenon requires protein synthesis and mTOR activity, and they should be cited and discussed in a resubmission*.

Thank you for your critical comment. We agree with this. We have now cited these papers in the references and rewrote the text of the Introduction and Discussion accordingly.